# Narrowing the A1c gap: Personalized modeling of HbA1c– continuous glucose monitor discordance in type 1 diabetes

Simon Lebech Cichosz[1]*, Camilla Heisel Nyholm Thomsen[1], David C. Klonoff[2],
Irl B. Hirsch[3], Morten Hasselstrøm Jensen[1,4]

**1** Department of Health Science and Technology, Aalborg University, Denmark, **2** Diabetes Research Institute, Mills-Peninsula Medical Center, San Mateo, California, United States of America, **3** Division of Metabolism, Endocrinology and Nutrition, Department of Medicine, University of Washington, Seattle, Washington, United States of America, **4** Data Science, Novo Nordisk, Søborg, Denmark

\* simcich@hst.aau.dk

## Abstract

This study aims to characterize the temporal discordance between CGM-derived glucose exposure and HbA1c over time in individuals with type 1 diabetes, and to explore the development of a statistical model to adjust the relationship between these measures based on previously observed individual discrepancies. We paired CGM-data in a 60-day window prior to each HbA1c measurement and included individuals with type 1 diabetes with multiple pairs to assess and model discordance over time. Discordance was defined as difference between HbA1c and Glucose Management Indicator at each pair. At baseline (first pair), participants were categorized into three groups based on the degree of discordance: positive (≥0.5%), negative (≤–0.5%), and neutral (within ±0.5%). A multiple linear regression model incorporating historical discordance values, HbA1c levels, and the current GMI was utilized for an adjustment. 477 individuals were included and 1,523 instances of paired HbA1c and CGM-data were analyzed. Absolute discordance of ≥0.5% was observed in 31% of cases. In 51% of instances, the direction of discordance in each pair was maintained. In the modeling analysis, GMI accounted for 69% of the variance in HbA1c levels (r = 0.83, p < 0.001, MAE = 0.42%). Adjusting improved variance explainability to 82% (r = 0.90, p < 0.001, MAE = 0.33%). HbA1c-CGM discordance is highly prevalent, and while inter-individual discordance shows some degree of persistence, it also appears to vary over time for a substantial proportion of individuals. Adjusting for individual discordance in the short term can improve the alignment between adjusted GMI and laboratory-measured HbA1c.

which permits unrestricted use, distribution, and reproduction in any medium, provided the original author and source are credited.

**Data availability statement:** The data used in this study were obtained from the T1DiabetesGranada and REPLACE-BG trials. Access to the T1DiabetesGranada dataset is subject to restrictions and was granted specifically for this research. Researchers interested in accessing this dataset should contact the original data custodians; further information is available: http://doi.org/10.1038/s41597-023-02737-4 . Access to the REPLACE-BG trial data can be requested through the JAEB Center for Health Research: https://public.jaeb.org.

**Funding:** This work was supported by i-SENS, inc. (www.i-sens.com) (approx. 2.1 MDKK to SLC). The funders had no role in study design, data collection and analysis, decision to publish, or preparation of the manuscript.

**Competing interests:** I have read the journal's policy and the authors of this manuscript have the following competing interests:The research was funded by i-SENS, Inc (Seoul, South Korea) and SLC's involvement with the company did not influence the design, implementation, or interpretation of the study. SLC have received research funding from i-SENS, Inc (Seoul, South Korea), which manufactures some of the product types discussed in this paper. However, the study was conducted independently, and the authors declare that their involvement with i-SENS, Inc (Seoul, South Korea) did not influence the findings or conclusions of the study. DCK is a consultant to: Afon Technology, Atropos Health, embecta, GlucoTrack, Lifecare, Novo Nordisk, SynchNeuro, and Thirdwayv.

## Author summary

The management of type 1 diabetes relies on two key tools: continuous glucose monitoring (CGM) and laboratory-measured HbA1c. While both measure sugar levels, they often disagree, leading to a "discordance" where a patient's CGM-calculated average does not match their clinical blood test.

We found that clinically significant discordance is common, affecting 31% of cases. Importantly, while this discrepancy tends to persist in the short term, it is not permanent and can vary over longer periods, suggesting it is influenced by transient factors like behavior or biology rather than genetics alone. To address this, we developed a personalized statistical model that uses an individual's historical data to "adjust" the CGM estimate. This adjusted GMI significantly improved the alignment with laboratory results. These findings provide a practical method for clinicians to better interpret glucose data, ensuring more precise and personalized care for people living with diabetes.

## Introduction

In recent decades, continuous glucose monitoring (CGM) and glycated hemoglobin (HbA1c) have emerged as cornerstones in the management of diabetes, each providing crucial yet distinct insights into glycemic control. While CGM devices offer real-time data on interstitial glucose levels and facilitate dynamic tracking of glycemic variability throughout the day, HbA1c remains the standard for assessing long-term average glucose control over a period of approximately two to three months [1]. Large population-based studies evaluating patient prognosis over extended periods have traditionally relied on HbA1c measurements [2,3]. In addition, HbA1c is not only straightforward to measure but also represents a cost-effective biomarker widely utilized across healthcare systems [4].

Despite their shared objective of quantifying glycemic exposure, mounting evidence reveals that HbA1c and CGM-derived measures of glucose exposure, such as the glucose management indicator (GMI) [5], often diverge in clinically meaningful ways. The relationship between such biomarkers is neither linear nor universally consistent; significant inter- and intra-individual discordances have been observed that challenge our understanding of glycemic dynamics and complicate clinical decision-making [6]. A study analyzing 641 individuals with type 1 and type 2 diabetes reported that clinically relevant discordance between CGM estimated HbA1c and laboratory-measured HbA1c ($\geq \pm 0.5\%$) was observed in 50% of cases [7], a finding echoed across multiple trials and real-world studies [8–11].

HbA1c, a reflection of cumulative glycemic exposure, encapsulates not only the biochemical process of glycation but also physiological factors such as red blood cell turnover, individual differences in glycation rates, and potential variabilities in

erythrocyte lifespan [12–14]. These factors can contribute to discordance between the mean glucose levels recorded by CGM and the HbA1c values measured in clinical settings. For instance, two individuals with similar CGM profiles may exhibit markedly different HbA1c readings, raising concerns about over- or underestimation of glycemic exposure when relying solely on HbA1c for clinical decisions [15–17].

Some studies have proposed that genetic or epigenetic factors may also influence the glycation process underlying HbA1c formation [18,19]. Furthermore, variations in the glycation ratio and the hemoglobin glycation index (HGI) [20] - the relationship between mean glucose and HbA1c - have been associated with potential adverse clinical outcome [15,21–23]. Individuals with consistently elevated HbA1c despite comparable glucose levels may be considered "high glycators," potentially facing greater microvascular risk [6,24,25]. Conversely, "low glycators" may face undertreatment when decisions are based solely on HbA1c. These observations underline the clinical importance of identifying and understanding factors driving the observed discordance.

Although several studies have identified discordance between CGM-derived measures of glucose exposure and HbA1c values [26–32], the underlying associations remain inadequately characterized. A recent analysis of data from the GOLD and SILVER trials [33] demonstrated significant inter-individual variability in HbA1c relative to both mean glucose levels and time in range (TIR), with these deviations persisting over time to some extent. However, it remains unclear how these observed inter- and intra-individual differences in discordance can be leveraged to develop improved models that correlate CGM data with HbA1c, especially important as clinical decisions today rely on smaller and smaller differences in HbA1c [19,34]. Enhancing this concordance could potentially provide a more precise assessment of individual glycemic risk, thereby informing clinical decision-making and the evaluation of therapeutic interventions - particularly in scenarios where simultaneous measurements across both modalities are not feasible or consistent.

This study aims to characterize the temporal discordance between CGM-derived glucose exposure and HbA1c in individuals with type 1 diabetes over time, and to explore the development of a statistical model to adjust the relationship between these measures based on previously observed discrepancies.

## Methods

The study design includes a characterization of temporal discordance in individuals with diabetes using long-term CGM data with multiple associated HbA1c measurements and an exploration of a modeling strategy for minimizing this discordance. A schematic overview of the methodological framework - including data preprocessing, analytical techniques, and modeling approach - is presented in Fig 1.

### Study data

This analysis utilized data from two studies on individuals with type 1 diabetes.

Cohort A: The T1DiabetesGranada study [35], collected CGM over a four-year period from 736 individuals with type 1 diabetes (T1D) residing in Granada, Spain. The primary device employed throughout the study was the FreeStyle Libre 2, though the initial phase involved the use of the first-generation FreeStyle Libre device. Biochemical parameters, including HbA1c, were scheduled for collection every three to six months, depending on the assessment and recommendations of the patient's physician.

Cohort B: Data from the REPLACE-BG trial [36]. The study was designed as a 6-month, multicenter, parallel-group, randomized clinical trial. The primary objective was to evaluate the safety and efficacy of routine CGM use without confirmatory blood glucose testing. Eligible participants were adults with type 1 diabetes using insulin pump therapy with a HbA1c level below 8.5% and no history of severe hypoglycemia or hypoglycemia unawareness. A total of 225 participants were enrolled and used CGM (Dexcom G4) for a period of up to six months. HbA1c measurements were collected at baseline, 13 weeks, and 26 weeks.

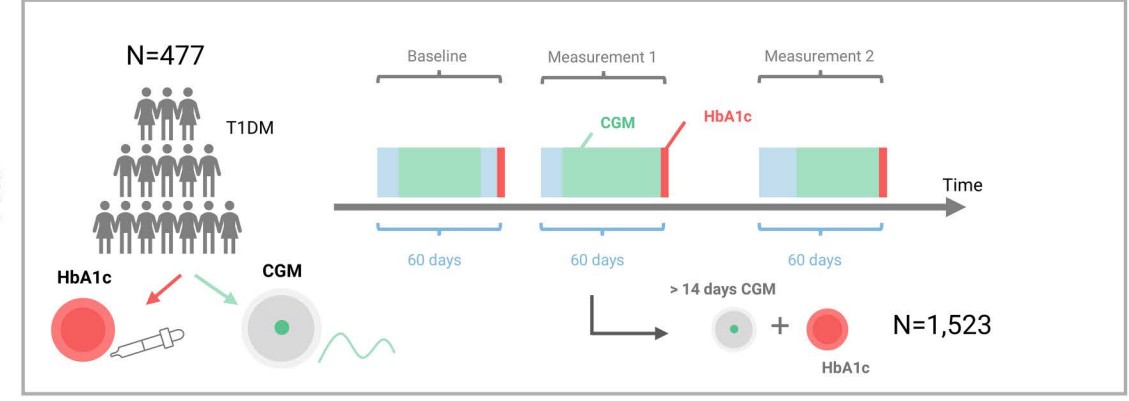

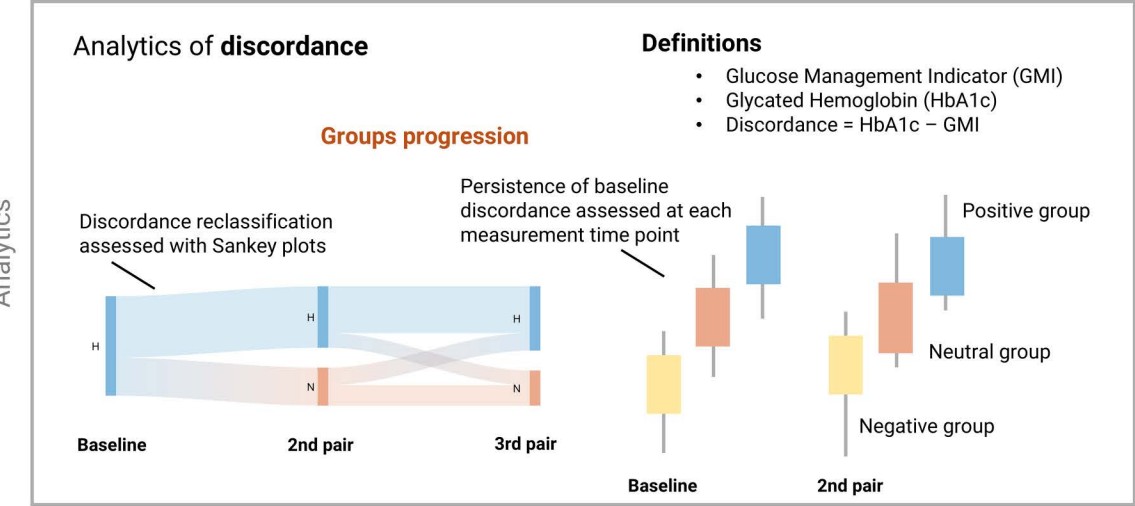

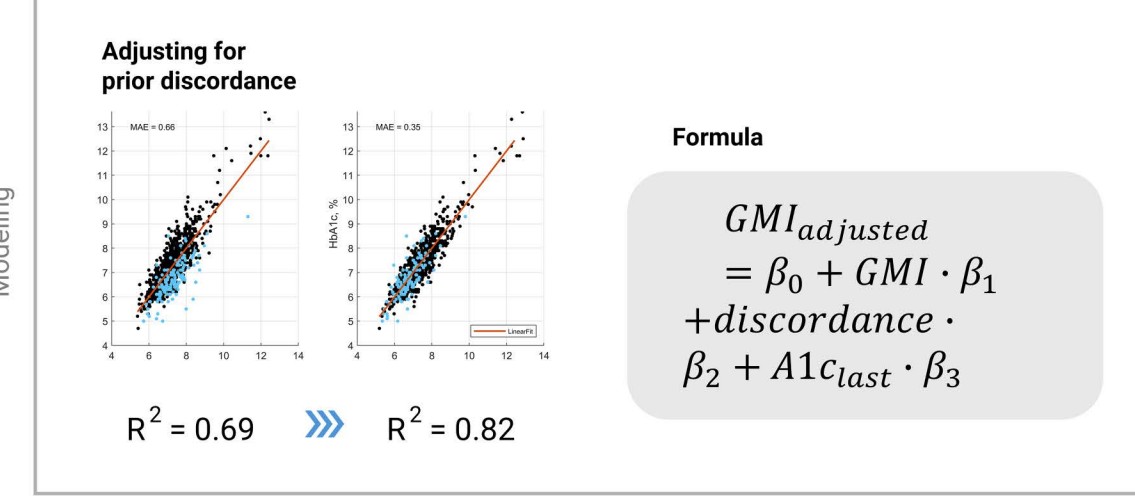

**Fig 1. Overview of the study approach. (Data):** Raw data were preprocessed to generate paired HbA1c measurements and corresponding CGM data from a defined period preceding each HbA1c test. **(Analytics):** Discordance between HbA1c and CGM-derived metrics was quantified for each pair and analyzed longitudinally. Individuals were grouped based on baseline discordance, and their trajectories were followed through the second and third measurement pairs. **(Modeling):** A statistical model was developed to adjust the Glucose Management Indicator (GMI) based on previously observed discordance and HbA1c levels.

## Preprocessing

For this analysis, patient data were evaluated from both the randomized trial (intervention and control groups; Cohort B) and an observational study (Cohort A). Inclusion criteria for the current analysis required patients to have instances of at least 14 days of active CGM use within the 60-day period preceding a laboratory-measured HbA1c and a minimum of two HbA1c measurements with such paired CGM. This approach has also been employed in previous research [33]. An overview of the data selection process is illustrated in Fig 1 ("Data").

All raw glucose traces were extracted at their native sampling frequency (5- or 15-minute intervals), and duplicate entries were removed. Periods affected by sensor dropout or physiologically implausible glucose values (<40 mg/dL or >400 mg/dL) were excluded. To avoid introducing bias, no temporal interpolation was applied [37]. CGM data cleaning and computation of paired metrics were performed using specialized software designed for standardized processing of glucose time-series data [38].

## Discordance assessment

The analysis aimed to assess the association between the Glucose Management Indicator (GMI) [5] and laboratory-measured HbA1c, using paired GMI–HbA1c values derived from the combined data sources (Cohorts A and B). GMI is a metric derived from the mean glucose measured by CGM and serves as a standardized estimate to facilitate direct comparison with HbA1c.

Discordance formula:

$$GMI(\%) = 3.31 + 0.02392 \frac{1}{N} \sum_{i \in ind} CGM_i \, [\, mg/dL]$$
$$Discordance = A1c - GMI$$

We examined whether individual-level discordance in the HbA1c–GMI relationship persisted over time, aiming to identify participants who consistently deviated. Clinically relevant discordance was defined as an absolute difference of ≥0.5% between GMI and HbA1c. The proportions of participants exceeding these thresholds were calculated from the data. At baseline (first GMI – HbA1c pair), participants were categorized into three groups based on the degree of discordance: positive discordance (≥0.5%), negative discordance (≤–0.5%), and neutral (within ±0.5%). This classification enabled the assessment of discordance persistence across subsequent HbA1c– GMI pairs (second and third). Additionally, to evaluate whether discordance persisted over time, we examined the relationship between the time interval from baseline to each subsequent measurement pair and the corresponding absolute discordance magnitude. For each individual, time intervals were calculated as the number of days between the baseline assessment and each follow-up measurement. Pearson correlation analysis was then performed to quantify the association between time interval and absolute discordance. This approach allowed us to assess whether longer intervals were associated with attenuation or persistence of discordance over time.

## Modeling and assessment approach

We hypothesized that discordance between estimated and laboratory-measured HbA1c exhibits a substantial, linearly modellable influence on future discordance. To test this, we employed a multiple linear regression model incorporating individual prior discordance value, last (prior) HbA1c level, and the current calculated GMI (independent variables) to predict the current HbA1c level (dependent variable). Use of a linear multi-parameter regression model was motivated by its interpretability and the ability to clearly quantify the contribution of individual predictors: an aspect particularly relevant for clinical understanding and adoption.

This model aimed to generate an individualized, adjusted GMI value, which we hypothesized would more closely approximate the patient's current laboratory-measured HbA1c.

Additionally, we explored whether including demographic variables - specifically age and gender -as independent predictors could further improve the accuracy of the adjusted estimates.

Additionally, we conducted supplementary analyses using a generalized additive model (GAM) with 5-fold cross-validation to assess potential nonlinear relationships. The GAM included the same dependent and independent variables as the linear model, and its performance was directly compared with that of the linear model.

## Statistical analysis

Descriptive statistics were calculated for demographic and clinical characteristics. Continuous variables are reported as mean ± standard deviation (SD) or median with interquartile range (IQR), as appropriate. Categorical variables are presented as counts and percentages.

Model performance was evaluated using the Pearson correlation coefficient, the coefficient of determination ($R^2$), and the mean absolute error (MAE) between predicted and observed HbA1c values. As a sensitivity analysis, we conducted evaluations across the full dataset as well as within subgroups characterized by clinically relevant positive and negative discordance. For comparisons of predictive performance between models, dependent correlation coefficients derived from the same sample were statistically compared using the Meng, Rosenthal, and Rubin test. This test accounts for the dependency between correlations arising from the shared outcome variable and provides a p-value to assess significance. To evaluate differences in central tendency between measurements across discordance groups classified at baseline pair, the Kruskal–Wallis H test was used for comparisons of median discordance values at each pair (baseline, 2nd, 3rd). This nonparametric test accounts for the non-normal distribution of discordance measures. All statistical analyses were performed in MATLAB R2025a, and p-value < 0.05 was considered statistically significant.

## Ethics statement

The presented study is a reanalysis of existing and anonymized data from the T1DiabetesGranada/ REPLACE-BG clinical trials. The presented study in this paper did not need any approval form institutional and/or licensing committee, cf. Danish law on "Bekendtgørelse af lov om videnskabsetisk behandling af sundhedsvidenskabelige forskningsprojekter og sundheds-datavidenskabelige forskningsprojekter" (Komitéloven, kap. 4, § 14, stk. 3). We confirm that all methods were carried out in accordance with relevant guidelines and regulations. The original REPLACE-BG protocols and informed consent forms were approved by the institutional review board. Written informed consent was obtained from each participant prior to enrollment. An independent data and safety monitoring board provided trial oversight reviewing unmasked safety data during the conduct of the study. The T1DiabetesGranada study was reviewed and approved by the Ethics Committee of Biomedical Research of the Province of Granada (CEIm/CEI GRANADA), protocol code K134665CRL, ethics portal code 0698-N-21.

## Results

A total of 477 individuals with type 1 diabetes were included in the study. Across these participants, 1,523 instances of paired HbA1c measurements and corresponding CGM data were analyzed. The mean age of participants was 40 ± 16 years, and 274 of 477 (57%) were female. During the analyzed CGM periods, the mean HbA1c was 7.3 ± 1.0% (56 ± 11 mmol/mol), and the corresponding mean glucose was 163 ± 31 mg/dL (9.0 ± 1.7 mmol/L). The median duration of CGM data available for each HbA1c pair was 53.7 days (interquartile range: 38.5–58.0 days). The median time between cases with paired HbA1c measurements was 126 days (interquartile range: 90–196 days).

## Discordance

Among all paired measurements, an absolute discordance of ≥0.5% and ≥1% was observed in 31% and 7% of cases, respectively. Overall reclassification (positive or negative) rate from one measurement pair to the next across all available data was 49%. Similar prevalences were observed across analytical cohorts A and B; detasils are reported in S1 Text.

Participants classified as positive, neutral, or negative discordance at baseline continued to exhibit significantly different discordance values at subsequent visits. This was supported by a highly significant Kruskal–Wallis test (p < 0.001) and confirmed by post-hoc pairwise comparisons between all groups (all, p < 0.001), demonstrating stability of the discordance phenotype.), Fig 2. A Sankey diagram, Fig 3, show specifically how the group, positive/negative, trajectory from baseline to the 2nd and 3rd paired measurements – this indicates that a proportion (42% for positive group/ 47% for negative group) of individuals remain in their initial group. We observed similar trends across both analytical cohorts, as presented in S1-S4 Figs and S1 Text.

Furthermore, correlation analysis between the time interval from baseline and absolute discordance magnitude revealed a negative correlation (r = -0.17, p < 0.006). These findings suggest that while discordance may exhibit short-term persistence, it is not necessarily stable over longer time intervals for all individuals.

### Modeling

In the modeling analysis, the GMI accounted for 69% of the variance in HbA1c levels (r = 0.83, p < 0.001), with a mean absolute error (MAE) of 0.42%. Incorporating the most recent observed discordance and HbA1c as additional predictors improved the model's performance (p < 0.001), explaining 82% of the variance (r = 0.90, p < 0.001) and reducing the MAE to 0.33%. The inclusion of demographic variables such as age and gender did not contribute further to the explanatory power of the adjusted GMI model. Adjusting for CGM model resulted in a marginal improvement in performance ($R^2 = 0.82$, r = 0.91, MAE = 0.32%, p < 0.001); however, to enhance generalizability beyond the specific sensors investigated, this adjustment was not included in the final model. Fig 4 presents scatter plots comparing measured HbA1c with both GMI and adjusted GMI.

Focusing specifically on the positive discordance group, the unadjusted model yielded an MEA of 0.57%, which decreased to 0.40% following adjustment for the most recent discordance and HbA1c (p < 0.001). In the negative

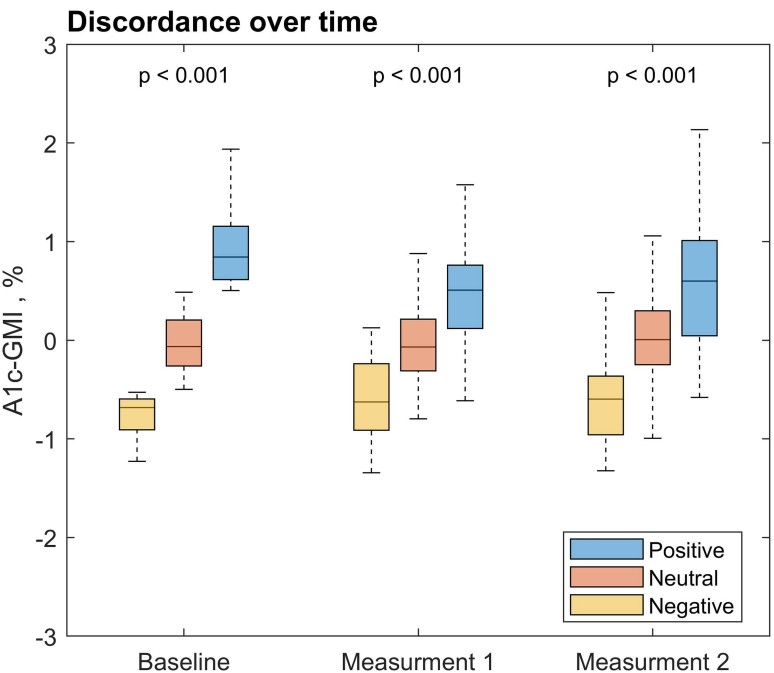

**Fig 2. Discordance groups over time.** The development in discordance in the groups – positive (discordance ≥0.5), neutral (-0.5 > discordance<0.5) and negative (-0.5≤) - from baseline to the first and second measurement.

**Group progression**

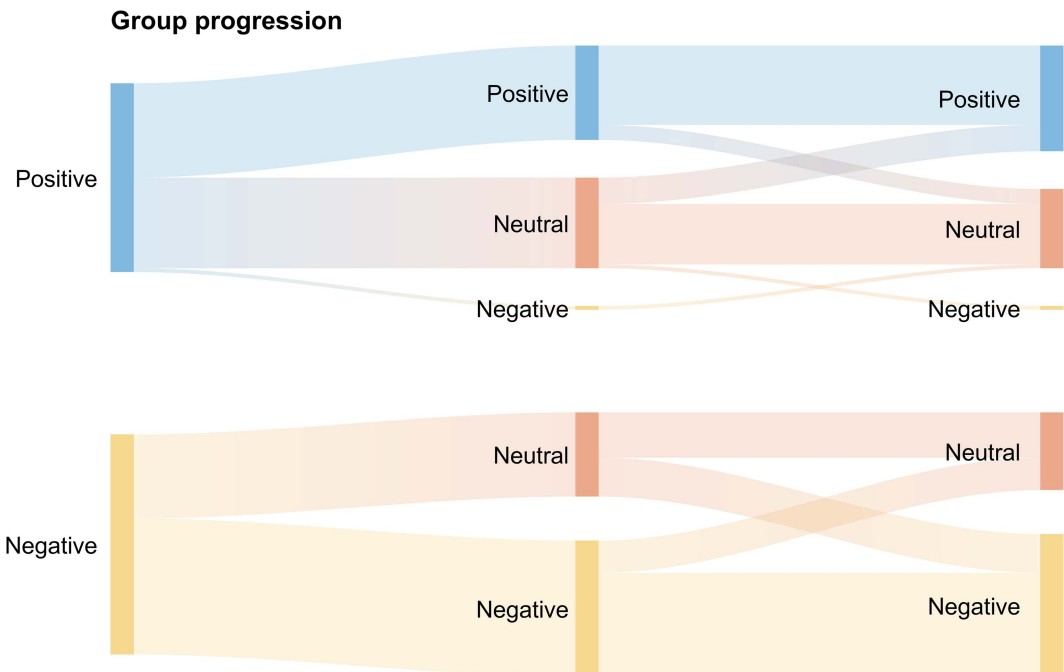

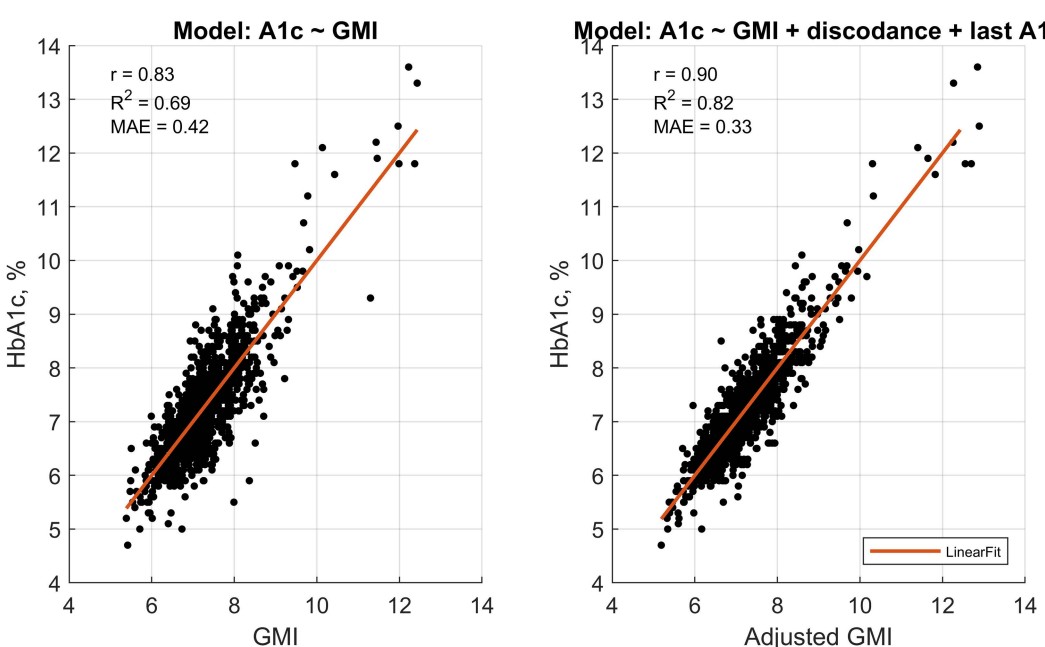

**Fig 3. Discordance group progression over time.** Sankey plots show the progression from the baseline discordance groups (left) – positive and negative – over time to measurement 1 (center) and measurement 2 (right).

**All observations**

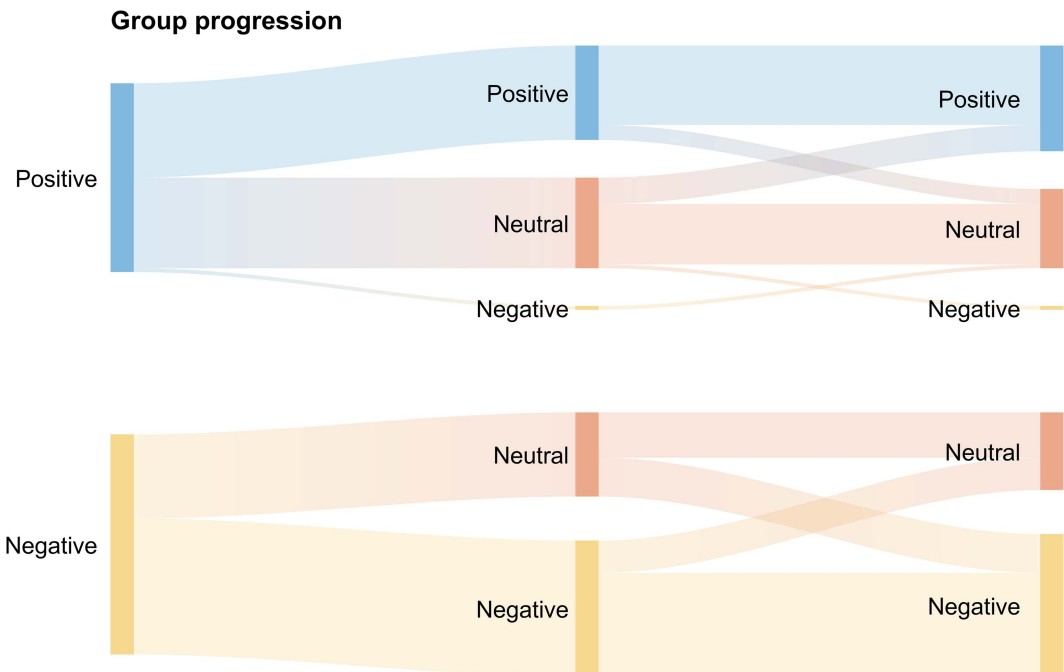

**Fig 4. Comparison of unadjusted and adjusted models.** Scatter plots display the relationship between HbA1c and GMI before (unadjusted) and after (adjusted) model calibration. Each plot includes the Pearson correlation coefficient (r), coefficient of determination ($R^2$), and mean absolute error (MAE) to assess model performance.

discordance group, the MEA was reduced from 0.66% to 0.35% after similar adjustment (p < 0.001). Fig 5 displays the corresponding scatter plots for the positive and negative discordance groups, illustrating the relationship between measured HbA1c and both unadjusted and adjusted GMI values. The following formula was used to calculate the adjusted GMI values:

Adjusted GMI:

$$aGMI\,(\%) = \beta_0 + GMI{\cdot}\beta_1 + prior\ Discordance{\cdot}\beta_2 + last\ HbA1c{\cdot}\beta_3$$

$$\beta_0 = -0.447$$
$$\beta_1 = 0.913$$
$$\beta_2 = 0.519$$
$$\beta_3 = 0.148$$

Overall, the GAM approach (non-linear) performed slightly worse than the linear model. Across the full cohort, the cross-validated $R^2$ for the GAM was 0.78, compared with 0.82 for the linear model. Similarly, the Pearson correlation between observed and predicted HbA1c was r = 0.88 for the GAM versus r = 0.90 for the linear model (p < 0.05). These results suggest that incorporating nonlinear effects via GAM did not substantially improve predictive performance in this dataset. The linear regression model therefore provides a simpler and more interpretable approach for predicting HbA1c based on GMI, prior discordance, and recent HbA1c.

## Discussion

In this study, utilizing data from two clinical trials comprising 477 individuals with long-term CGM and multiple paired HbA1c measurements, we observed both inter- and intra-individual discordance between laboratory-measured HbA1c and the GMI derived from CGM data. These inter-individual differences were clinically meaningful, with substantial HbA1c deviations (>0.5%) identified in 31% of cases. Our findings indicate that discordance exhibited some degree of intra-individual persistence over time. However, for many individuals this discordance does not seem stable over longer time-spans. Incorporating prior knowledge of individual discordance patterns enabled the development of an adjusted GMI, which improved alignment with observed HbA1c values. The addition of demographic variables, such as age and gender, did not further enhance the model's predictive performance.

Previous studies examining the relationship between mean glucose levels from CGM and HbA1c have reported correlation coefficients ranging from r = 0.73 to 0.80 [29,31,39,40], consistent with our findings. However, by incorporating prior discordance into our model, we were able to improve the correlation coefficient and explained variance in HbA1c from 69% to 82%, while also significantly reducing the mean absolute error. A recent study by Isaksson et al. [33] reported persistence in discordance over the duration of the trial; our results build upon this by demonstrating that while discordance may persist short term, it does not appear to be stable over longer periods. In both the positive (≥0.5%) and negative (≤-0.5%) discordance groups, the magnitude of discordance slowly declined over time. This trend suggests that discordance may, for some individuals, be influenced more by modifiable or transient factors - such as behavioral, hormonal or psychological variables - rather than by fixed inter-individual characteristics like genetics. For instance, menopause in women has been associated with elevated HbA1c levels independent of actual glucose concentrations, potentially because of hormonal changes and altered red blood cell turnover [41]. Also, CGM coverage appears to be linked to adherence. As recently demonstrated by Cichosz et al., days with lower CGM coverage were associated with reduced inter-individual time in range in a cohort of 97,000 participants, encompassing both clinical study and real-world data from individuals with type 1 and type 2 diabetes [42]. Individual variability in red blood cell (RBC) lifespan may represent an etiology of glycemic discordance, as highlighted by recent work from Xu et al [43]. Recently, Cohen et al. presented

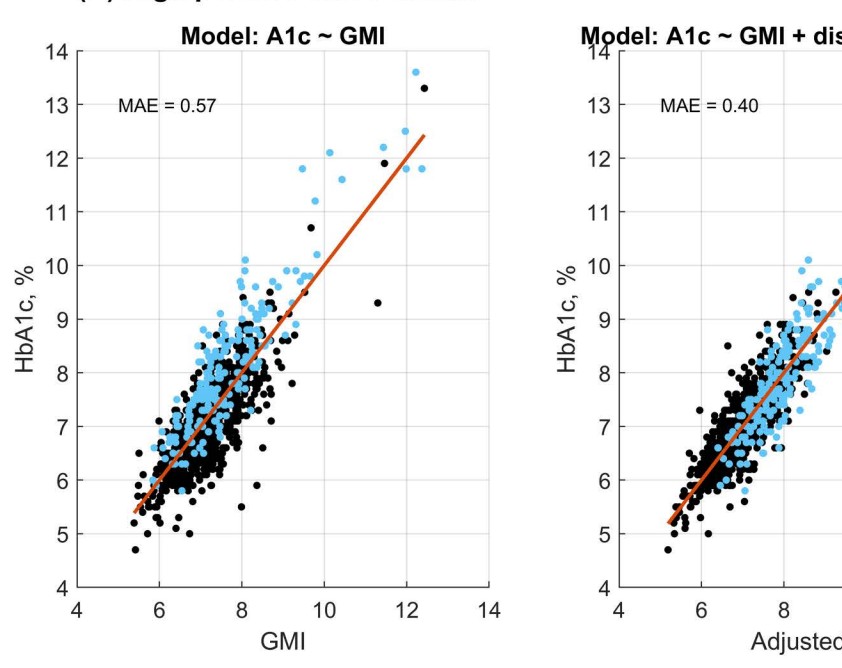

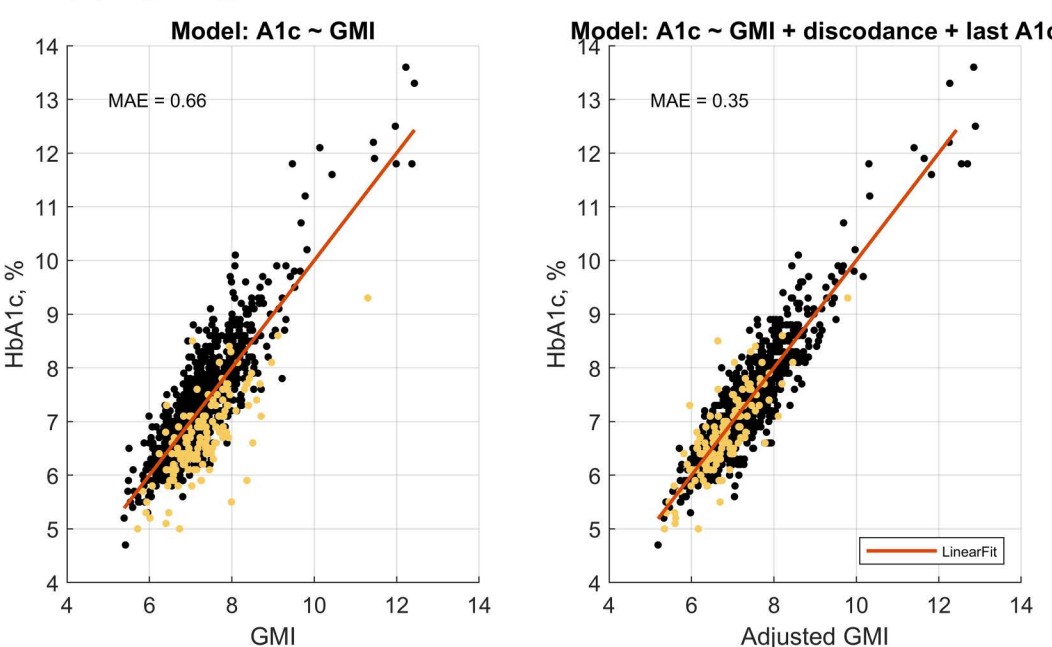

**Fig 5. Subgroup comparison of adjusted and unadjusted models.** Performance of the unadjusted and adjusted models is compared separately for (A) individuals with prior positive discordance and (B) individuals with prior negative discordance. Mean absolute error (MAE) is reported for each subgroup to illustrate improvements in predictive accuracy following adjustment.

initial findings on adjusting for these individual differences in lifespan [44]. Additionally, sensor accuracy may contribute to observed discordance, as performance can vary both between CGM devices and within individuals over time. Supporting this, Freckmann et al. [45] demonstrated significant variability across different CGM systems in comparative testing.

## Implications

The findings from this study, particularly regarding the temporal dynamics of discordance and the proposed adjustment to GMI, have several clinical and research implications. First, in cases where substantial discordance is observed in clinical practice, further investigation may be warranted to identify underlying causes and if persistence exists for the individual - especially physiological conditions such as iron deficiency or menopause that can affect HbA1c. Second, recognizing that discordance may persist temporarily but is unlikely to be permanent, in general, is important when evaluating CGM and HbA1c in therapeutic decision-making. However, further investigation into the discordance between HbA1c and GMI with newer sensors (such as the G7, Simplera, and Libre 3 [45,46]), particularly the impact of sensor switching on this discordance, is warranted. Thirdly, the proposed model offers a straightforward method for adjusting GMI in the absence of a paired HbA1c measurement, which can enhance its utility in routine clinical practice.

Finally, our findings contribute to the growing body of evidence highlighting substantial clinical discordance between the GMI and HbA1c across various diabetes subpopulations. Recent work by Elizabeth Selvin [32] has underscored that GMI and HbA1c are not consistently correlated at the individual level, prompting the argument that the GMI concept may have outlived its clinical utility. In light of our results and those of others, we advocate for a re-evaluation of the GMI formula, taking into account: (1) diabetes subtype, (2) type of CGM sensor used, and (3) clinical and biological confounders that may influence HbA1c and potentially CGM-derived metrics. These often-overlooked factors must be considered to develop either a universally applicable or phenotype-specific GMI. While our present study illustrates the potential for such tailored adjustments, it should not be interpreted as a definitive or universal solution.

To date, HbA1c remains the standard reference metric for assessing glycemic control in individuals with diabetes mellitus, primarily because of its established association with the risk of microvascular and long-term macrovascular complications [47]. However, this assumption has been called into question, with emerging evidence suggesting limitations in its reliability across diverse patient populations and clinical contexts [48]. HbA1c is affected by multiple non-glycemic factors that can cause substantial inter-individual variation in glycation, highlighting that HbA1c can misrepresent a patient's true glycemic exposure. While CGM-derived metrics such as time-in-range (TIR), glycemic variability, the Glycemia Risk Index and other may provide superior insights into therapeutic efficacy [40,49] and may also serve as stronger predictors of diabetes-related complications [50,51], access to continuous CGM remains limited in some groups of diabetes and regions [52], particularly among individuals with type 2 diabetes. For many in this population, glycemic control is still predominantly assessed via HbA1c. In this context, intermittent CGM use can be a valuable strategy, particularly in the early stages following a type 2 diabetes diagnosis, to establish individualized glycemic profiles that can inform treatment decisions and monitor disease progression [53]. Moreover, intermittent CGM use can enhance patient understanding of their condition and support insulin titration when needed [53]. For individuals with limited or occasional CGM access, understanding and adjusting for discordance between CGM-derived metrics and HbA1c is clinically important to ensure accurate interpretation and to guide optimal therapeutic interventions.

## Strengths and limitations

A key strength of this study is the inclusion of a large, heterogeneous cohort of individuals with type 1 diabetes, each contributing multiple paired CGM and HbA1c measurements. The extended follow-up period of up to four years further allowed for the assessment of sustained patterns of CGM-HbA1c discordance over time. However, this study has several limitations. We lacked sufficient data on ethnic differences, conditions such as anemia and other blood disorders, which are

known to potentially influence HbA1c levels. Additionally, detailed information on individual treatment regimen and temporal changes in therapy during the observation period was not collected, precluding an evaluation of their potential impact on glycemic metrics. Paired CGM profiles were included for each HbA1c value if at least 14 days of CGM data were available within the 60 days preceding the HbA1c measurement. Although CGM data availability was generally high, the selected time window may not fully capture long-term glycemic patterns represented by HbA1c, particularly in patients with unstable glucose control. This methodological constraint may have led to underrepresentation of true long-term glycemic status in some cases. Although we observed similar trends across both cohorts using two different glucose sensors, advances in sensor technology have generally reduced the error between interstitial and blood glucose measurements. A future longitudinal study investigating discordance with modern sensors is needed to better understand the variability in discordance.

## Conclusions

HbA1c-CGM discordance is highly prevalent, and while inter-individual discordance shows some degree of persistence with this older generation of sensors, it also appears to vary over time for a substantial proportion of individuals. Adjusting for individual discordance in the short term may improve the alignment between adjusted GMI and laboratory-measured HbA1c. This approach could be particularly valuable for individuals with diabetes who have limited or intermittent access to CGM, or in clinical settings where paired CGM and HbA1c measurements are not consistently available for longitudinal assessment of glycemic control. Nonetheless, there is a need to revisit the GMI formula with consideration of additional influencing factors.

## Supporting information

**S1 Text. Additional individual analytics for each data source included in the main analysis.**
(DOCX)

**S1 Fig. Discordance group progression over time (Cohort A/ The T1DiabetesGranada study).** Sankey plots show the progression from the baseline discordance groups (left) – positive and negative – over time to measurement 1 (center) and measurement 2 (right).
(DOCX)

**S2 Fig. Discordance group progression over time (Cohort B/ The REPLACE-BG trial).** Sankey plots show the progression from the baseline discordance groups (left) – positive and negative – over time to measurement 1 (center) and measurement 2 (right).
(DOCX)

**S3 Fig. Discordance groups over time (Cohort A/ The T1DiabetesGranada study).** The development in discordance in the groups – positive (discordance ≥0.5), neutral (-0.5 > discordance<0.5) and negative (-0.5≤) - from baseline to the first and second measurement.
(DOCX)

**S4 Fig. Discordance groups over time (Cohort B/ The REPLACE-BG trial).** The development in discordance in the groups – positive (discordance ≥0.5), neutral (-0.5 > discordance<0.5) and negative (-0.5≤) - from baseline to the first and second measurement.
(DOCX)

## Acknowledgments

**Disclaimer:** The source of the data is from the T1DiabetesGranada/ REPLACE-BG trial, but the analyses, content and conclusions presented herein are solely the responsibility of the authors and have not been reviewed or approved by trial group(s).

## Author contributions

**Conceptualization:** SL Cichosz, Camilla Heisel Nyholm Thomsen, David C. Klonoff, Irl B. Hirsch.

**Data curation:** SL Cichosz.

**Formal analysis:** SL Cichosz.

**Funding acquisition:** SL Cichosz.

**Investigation:** SL Cichosz.

**Methodology:** SL Cichosz, Camilla Heisel Nyholm Thomsen, David C. Klonoff, Irl B. Hirsch, Morten Hasselstrøm Jensen.

**Project administration:** SL Cichosz.

**Resources:** Morten Hasselstrøm Jensen.

**Software:** SL Cichosz.

**Supervision:** Irl B. Hirsch, Morten Hasselstrøm Jensen.

**Visualization:** Camilla Heisel Nyholm Thomsen.

**Writing – original draft:** SL Cichosz.

**Writing – review & editing:** Camilla Heisel Nyholm Thomsen, David C. Klonoff, Irl B. Hirsch, Morten Hasselstrøm Jensen.

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
