## [Decision Letter · Decision Letter 0]

3 Dec 2025

Response to Reviewers
Revised Manuscript with Track Changes
Manuscript
**Journal Requirements:**

Please send a completed 'Competing Interests' statement, including any COIs declared by your co-authors. If you have no competing interests to declare, please state "The authors have declared that no competing interests exist". Otherwise please declare all competing interests beginning with the statement "I have read the journal's policy and the authors of this manuscript have the following competing interests:"

Please amend your detailed Financial Disclosure statement. This is published with the article. It must therefore be completed in full sentences and contain the exact wording you wish to be published.a. State the initials, alongside each funding source, of each author to receive each grant. For example: "This work was supported by the National Institutes of Health (####### to AM; ###### to CJ) and the National Science Foundation (###### to AM)."

Please upload all main figures as separate Figure files in .tif or .eps format only. For more information about how to convert and format your figure files please see our guidelines:https://journals.plos.org/digitalhealth/s/figures

We have noticed that you have uploaded Supporting Information files, but you have not included a list of legends. Please add a full list of legends for your Supporting Information files after the references list.

**Additional Editor Comments (if provided):**
**Reviewers' Comments:**

**Comments to the Author**

1. Does this manuscript meet PLOS Digital Health’s publication criteria?

Reviewer #1: No

Reviewer #2: Partly

Reviewer #3: Yes

2. Has the statistical analysis been performed appropriately and rigorously?

Reviewer #1: No

Reviewer #2: Yes

Reviewer #3: Yes

3. Have the authors made all data underlying the findings in their manuscript fully available (please refer to the Data Availability Statement at the start of the manuscript PDF file)?

Reviewer #1: Yes

Reviewer #2: Yes

Reviewer #3: Yes

4. Is the manuscript presented in an intelligible fashion and written in standard English?

Reviewer #1: No

Reviewer #2: Yes

Reviewer #3: Yes

Reviewer #1: The study addresses an interesting topic on HbA1c–GMI discordance in type 1 diabetes, but there are some limitations that affect its current suitability for publication. The novelty is limited, as similar analyses have been published before, and the study relies entirely on previously published datasets without any primary data collection, which reduces the strength and generalizability of the findings. The modeling approach also raises concerns because lack of clarity on predictor and outcome parameters, and the temporal claims are not fully supported, since within-subject changes over time are not robustly accounted for. Additionally, important methodological details, such as CGM data handling and HbA1c measurement protocols, detailed methodology, are missing, and the clinical applicability of the findings remains unclear. Addressing these points would substantially improve the study.

Reviewer #2: The paper is interestin and investigates a long standing problem, which not many people in the medical field understand. It is important and timely.

Regarding the model: the model is linear, based on multi-parameter regression. But the matter investigated may not be linear. Have the authors tried nonlinear methods for investigation?

Perhaps a nonlinear regression may address even better.

Or another nonlinear simple model, such as Artificia Neural Network.

Reviewer #3: This study by Cichosz and colleagues aims to characterise the temporal discordance between CGM-derived glucose exposure (defined by GMI) and laboratory measured HbA1c in individuals with type 1 diabetes, and evaluates a statistical approach for adjusting GMI based on previously observed discrepancies. Across all CGM-HbA1c paired measurements, absolute discordance ≥0.5% was observed in 31% cases. In 51% of sequential pairs individuals maintained the same direction of discordance positive vs negative (ie. systematically higher or lower HbA1c relative to GMI) and a notable proportion preserved this across baseline, measurement 1 and measurement 2. However, discordance magnitude declined over time. To address this persistent (but variable) discordance the authors propose an individualised method to adjust GMI, using a multiple linear regression model (fixed effects only) incorporating current GMI with historical discordance values and most recent HbA1c. They find that incorporating these additional predictors improved the model’s performance overall, explaining 82% of the variance (r = 0.90, p < 0.001) and reducing the MAE to 0.33%, finding similar reductions in MEA by restricting to individuals with prior high positive discordance or prior high negative discordance.

This is a timely and relevant study addressing an important issue in CGM metrics used in clinical care. Overall, the manuscript is well written, the data presented are robust and the primary conclusions are supported by the analyses. Appropriate statistical analysis have been applied, although several methodological details would benefit from clarification, in addition to my other comments below. The comments below aim to strengthen the clarity, interpretability, and clinical framing of the manuscript.

Introduction

Minor:

• Lines 66 remove “individual involving”

• Lines 66-68 “A study analyzing 641 individuals…. reported that clinically relevant discordance between CGM estimated HbA1c and laboratory-measured HbA1c (≥ ±0.5%) was observed in 72% of cases”:This study actually reported 50% ≥±0.5%. I think the authors have summed 50% and 22% reported for >0.5 and >1% (however the 22% would be part of the 50% reported). From the original paper: “Only 11% of our patients had discordance <0.1%, while 50% and 22% had differences ≥0.5% and ≥1%, respectively.” See fig 3 in original report.

• Although cited elsewhere, Selvin et al 2024 earlier in the report may be beneficial as this contains the majority of studies comparing GMI and HbA1c concordance

Methods

• The group names of high and low discordance is confusing. The authors refer to high discordance (≥0.5%), low discordance (≤–0.5%) however then describe it as positive and negative discordance groups elsewhere ie. line 149 and Figure 5 with the text starting line 203 not reflecting this either for example. While I understand that you may mean systematically higher or lower HbA1c relative to GMI, when discussing directionality it is clearer to use Positive and Negative (and Neutral) for your groups because an absolute discordance of 0.5% is “high” regardless. Low suggests “a small amount of discordance”.

• Some restructuring of the methods section to include a statistical analyses section would be beneficial for clarity of methods and consequently results. Some points to include here but not limited to:

-What test was used to compare within groups? (and what groups are compare ie. within discordance phenotype across time or between discordance phenotype at each time point - See later comment on Figure 2 results.)

-Alpha level?

-What stats software was use for analysis?

-Can move modelling performance assessment criteria to here.

-Can outline here for eg use of Pearson correlation analysis for correlations (including your time and discordance magnitude correlation assessment).

• In Figure 1 “data” what is the blue part of the bar? This suggests that some part of the 60 days of data didn’t have CGM available systematically? Please describe in methods if so. Of note, as referenced to the GOLD and SILVER study applying a similar look back period: they outline that “The CGM systems used in the current study store up to 30 days of active CGM data. In the current analyses, CGM data from GOLD and SILVER trials comprising a minimum of 14 days of active CGM measurements within 60 days before laboratory HbA1c were included”….. in which they describe in their Figure 1 translates to a maximum of 30 days of active CGM data per 60 period. On line 170: The median duration of CGM data available for each HbA1c pair was 53.7 days (interquartile range: 38.5–58.0 days), suggests similarly that at least one of the systems used can’t store >30 days of data whereas the other can.

-This may be worth a comment to outline this as this means that GMI from one system will only ever be based on 30 days of data vs from another system could be based on >30-60 days of data.

-I would like to know the split of the studies in the finial inclusion and to know the split of devices ie. Libre vs Dexcom– this comment could extend to creation a table of characteristics see results section.

• Figure 1 would benefit with some additional labelling to make it understandable without having to read the entire paper since it is referenced in Methods.

-Would prefer positive/neutral/negative groups labelled see earlier comment.

-Include measurement 1 and 2 under the box plot clusters and the legend key of the baseline discordance group per colour?

-Some short text that describe the analytic methods section. For example: “Discordance reclassification assessed with Sankey plots” above the Sankey plot image, above box plots: “Persistence of baseline discordance direction assessed at each measurement time point”?

Minor

• Lines 123-125 describing inclusion criteria and Figure 1 don’t seem to align. Do you mean that, for a HbA1c measurement to be included it had to be preceded by at least 14 consecutive days of active CGM use within the prior 60 days? And participants needed more than one such CGM-HbA1c pair to contribute to the longitudinal analyses? If so could reword to something similar, otherwise the way it is currently worded seems to suggest that each 60-day period per HbA1c must contain multiple (at least) 14-day blocks -which doesn’t seem rational.

• Lines 149-150 you describe the method of the Pearson correlation analysis with time interval for to evaluate its influence on discordance persistence over time. You actually describe this clearer in the results as a correlation analysis between the time interval from baseline and absolute discordance magnitude. You use persistence when referencing to the reclassification of discordance direction ie. High-high-high which confuses the intial statement in the method. Consider rewording the methods sentence to reflect what you describe in the results.

• Add (r) after Pearson correlation coefficient

Results

Minor

• Does the 51% relate to figure 3? If so why is this not referenced here. If not then perhaps some clarification.

• Adding the percentages to the Sankey alluviums would be helpful in Suppl fig 1 and Figure 3.

• Lines 182-183 you state a proportion of individuals remained in their initial groups. Could you state the percentage. (above bullet may also help address this)

• Figure 2 would be easier to interpret if methods have more description. It would be unclear to the brief reader if comparing within baseline discordance phenotype across time ie all yellow boxes across time, or within each time point between all discordance groups. I am assuming that you are comparing the latter, ie between discordance phenotypes at each timepoint.

-Addition of the pvalue to the graph here above each time point may help. (as well as a sentence in statistical analyses methods paragraph that can better describe this and the tests you are using ie.: “Participants were classified at baseline into three discordance categories (Positve, Neutral, Negative) based on the baseline A1c–GMI difference. These baseline-defined groups were carried forward without reclassification. At Measurement 1 and Measurement 2, new discordance values (A1c–GMI) were calculated for each participant and compared across the three baseline groups. Because discordance was non-normally distributed (?), between-group differences at each timepoint were evaluated using the Kruskal-Wallis test with alpha = 0.05….” (replace with the test you used)

-You could also make the result clearer described ie.” A significant p-value at subsequent visits (p < 0.001) indicated that participants classified as Positive, Neutral, or Negative discordance at baseline continued to exhibit significantly different discordance values at subsequent visits, supporting stability of the discordance phenotype.”

• Supplementary S4 is mislabelled as S3.

• Could add “GMI” and “adjusted GMI” on the x axis on Fig 4 not just “Estimated” for clarity

• Your equation for GMI uses historic discordance as stated on lines 155-156. Perhaps adding the word “last” to the equation box would be helpful (like you have “last hba1c”).

• Was there a reason to not include the neutral group in the Sankey diagram on Fig 3 / suppl fig 1? They were present in the Fig 2 box plots

Discussion

• There been a (very recent) proposal of an updated GMI (uGMI) metric from Bergenstal et al (July 2025 ADA abstract OR165) which accounts for population-based red blood cell factors as well as its utility in Shah et al., 2025 Diabetes Care. Some comment on this upon revision would be welcomed.

• A further point I would appreciate adding to the discussion is that while the authors appropriately highlight the limitations of the current GMI formula (which was developed in old sensors and in a small, mostly white cohort), it is equally important to acknowledge that HbA1c itself is not an ideal reference standard. HbA1c remains the regulatory and clinical gold standard, yet it is influenced by multiple non-glycaemic factors which can cause substantial inter-individual variation in glycation and is a well known "lagging marker". As the authors note in their introduction, “two individuals with similar CGM profiles may exhibit markedly different HbA1c readings,” underscoring that HbA1c may misrepresent a patient’s true glycaemic exposure. This raises the question of whether adjusting GMI to more closely align with HbA1c truly resolves discordance, or whether both measures are partially flawed proxies of a more complex underlying physiology. The recent development of the population-adjusted uGMI (Bergenstal et al 2025 described above) and its demonstrated predictive superiority over HbA1c for incident retinopathy (Shah et al, 2025 Diabetes Care) further reinforce the need to reconsider whether HbA1c is the optimal anchor for assessing true glycaemic exposure. At the same time, the authors make a valid and important point regarding inequitable access to CGM. Many individuals with type 2 diabetes rely primarily on HbA1c, and others have only intermittent CGM use. In such contexts, understanding and adjusting for discordance between CGM-derived estimates and HbA1c is essential for safe interpretation and clinical decision-making. However, as CGM access expands and CGM-based biomarkers evolve, there may be a need to revisit whether HbA1c should continue to serve as the sole reference point for calibrating CGM-derived metrics such as GMI.

Minor

• Line 223-224 Also able to improve the correlation - could state that in here too since the previous sentence mentions correlation only.

• Lines 227-232 Discordance likely changes as mediated by things like CGM data quality (CGM wear adherence, compression lows etc) and short-term glycemic exposure (ie. if you have a recent reduction in your variability, GMI might be quicker to stabilise than hba1c). It is these things which are influenced by behavioural and physiological variables. The statement would benefit from that additional information.

• Line 227-232 Subtle, but the way this is worded suggests that menopause is a behavioural/psychological variable…. Please consider rewording or adding “hormonal” to the list.

• Lines 235-237 Clarity may be warranted here with perhaps addition of paper from Eichenlaub et al: Eichenlaub et al 2025 JDST found substantial differences in accuracy between CGM systems worn by the same individual. Freckmann et al 2025 Diabetes Care demonstrated that these differences in accuracy significantly affect the CGM derived metrics.

**Do you want your identity to be public for this peer review?** For information about this choice, including consent withdrawal, please see our Privacy Policy

Reviewer #1: No

Reviewer #2: **Yes:** Maia Angelova

Reviewer #3: No

**Figure resubmission:**

**Reproducibility:** To enhance the reproducibility of your results, we recommend that authors of applicable studies deposit laboratory protocols in protocols.io, where a protocol can be assigned its own identifier (DOI) such that it can be cited independently in the future. Additionally, PLOS ONE offers an option to publish peer-reviewed clinical study protocols. Read more information on sharing protocols at https://plos.org/protocols?utm_medium=editorial-email&utm_source=authorletters&utm_campaign=protocols

---

## [Decision Letter · Decision Letter 1]

23 Jan 2026

Narrowing the A1c gap: personalized modeling of HbA1c– continuous glucose monitor discordance in type 1 diabetes

PDIG-D-25-00341R1

Dear Professor Cichosz,

We are pleased to inform you that your manuscript 'Narrowing the A1c gap: personalized modeling of HbA1c– continuous glucose monitor discordance in type 1 diabetes' has been provisionally accepted for publication in PLOS Digital Health.

Best regards,

Krasimira Tsaneva-Atanasova

Academic Editor

PLOS Digital Health

**Additional Editor Comments (if provided):**

**Reviewer Comments (if any, and for reference):**

Reviewer's Responses to Questions

**Comments to the Author**

Reviewer #1: All comments have been addressed

Reviewer #2: All comments have been addressed

Reviewer #3: All comments have been addressed

publication criteria?

Reviewer #1: Yes

Reviewer #2: Yes

Reviewer #3: Yes

3. Has the statistical analysis been performed appropriately and rigorously?

Reviewer #1: Yes

Reviewer #2: Yes

Reviewer #3: Yes

4. Have the authors made all data underlying the findings in their manuscript fully available (please refer to the Data Availability Statement at the start of the manuscript PDF file)?

Reviewer #1: Yes

Reviewer #2: Yes

Reviewer #3: No

5. Is the manuscript presented in an intelligible fashion and written in standard English?

Reviewer #1: Yes

Reviewer #2: Yes

Reviewer #3: Yes

Reviewer #1: The revised version of the article may be considered for publication.

Reviewer #2: I would like to thank the authors for performing additional analysis using a non-linear model. While it is less accurate, it has most likely captured some non-linear features of the model, which may not as important as the linear characteristics.

Reviewer #3: The authors have addressed my comments. The revised manuscript is clear, well argued, and significantly improved.

**Do you want your identity to be public for this peer review?** For information about this choice, including consent withdrawal, please see our Privacy Policy

Reviewer #1: **Yes:** Dr. Arkaprabha Sau

Reviewer #2: **Yes:** Maia Angelova

Reviewer #3: No
